# 'Why Not This MAPF Plan Instead?'
# Contrastive Map-based Explanations for Optimal MAPF

**Martim Brandão, Yonathan Setiawan**

King's College London, UK

## Abstract

Multi-Agent Path Finding (MAPF) plans can be very complex to analyze and understand. Recent user studies have shown that explanations would be a welcome tool for MAPF practitioners and developers to better understand plans, as well as to tune map layouts and cost functions. In this paper we formulate two variants of an explanation problem in MAPF that we call contrastive "map-based explanation". The problem consists of answering the question "why don't agents A follow paths P' instead?"—by finding regions of the map that would have to be an obstacle in order for the expected plan to be optimal. We propose three different methods to compute these explanations, and evaluate them quantitatively on a set of benchmark problems that we make publicly available. Motivations for generating this type of explanation are discussed in the paper and include both user understanding of MAPF problems, and designer-aids to guide the improvement of map layouts.

## Introduction

Multi-Agent Path Finding (MAPF) plans are able to coordinate multiple agents to arrive at their destinations without colliding with obstacles or each other (Stern et al. 2019). These plans are complex to understand, analyze and tune (Brandao et al. 2022). For example, when used in warehouse automation or computer games, they typically involve hundreds of agents, as well as complex cost functions encoding multiple preferences. A recent user study (Brandao et al. 2022) has showed that MAPF experts and practitioners would be interested in having explainable planners in order to, for example, tune environment layouts and cost functions, and understand planner behavior. While current explanation-generation work in MAPF has focused on visualizing feasibility of plans (Almagor and Lahijanian 2020; Kottinger, Almagor, and Lahijanian 2021), practitioners are often more interested in asking contrastive questions about agent behavior, such as "why does agent A not take path X instead?" (Brandao et al. 2022).

In this paper we look at the problem of answering questions of the type "why don't agents $A'$ take paths $\pi'$ in the optimal plan?" where $A'$ could be both the set of all agents or a subset. We focus on map-based explanations for this type

of questions, as these have been reported to be useful to both warehouse layout designers (Brandao et al. 2022) and algorithm developers (Brandao et al. 2021a). The paper provides a formalization of the problem, three methods to solve it, and an open set of problems for benchmarking. Our approaches to solve the problem are inspired by a general search-based XAIP approach (Chakraborti et al. 2017), and multi-agent extensions of inverse shortest-paths (Ahuja and Orlin 2001; Brandao, Coles, and Magazzeni 2021). The main idea is to compute the smallest set of cells in the map that would have to be obstacles in order for the expected paths to become optimal. To the question "why don't agents A' take paths $\pi'$ in the optimal plan?", we therefore compute an explanation of the type "Because the cells/vertices X are free. These cells would have to be obstacles in order for $\pi'$ to be optimal". This paper is therefore an extension of recent work in map-based explanations for single-agent path finding (Brandao, Coles, and Magazzeni 2021).

Our contributions are the following:

1. We formulate two new explanation problems, called "full" and "partial" map-based MAPF explanations;
2. We develop and openly provide a new benchmark of problems for these types of explanations;
3. We propose and evaluate three methods for solving these problems: a search-based method which is optimal in both settings, a fast optimization-based method which is optimal in the "full" setting, and an incomplete agent-conflict-free method.

The paper is organized in the same order: it starts by introducing the problem, then the methods, and then results. These are followed by sections on related work and conclusions.

## MAPF Map-based Explanation Problems

Consider a MAPF problem on a graph $G = (V, E)$, with $M$ agents $A = \{a_1, ..., a_M\}$ whose starting vertices are given by $s : A \mapsto V$ and goal vertices $g : A \mapsto V$. At each timestep an agent can move to an adjacent vertex connected by an edge, or wait at the same vertex. Let $p_i(t)$ be the vertex where agent $a_i$ is located at time $t$. A path $p_i = [p_i(0), ..., p_i(T_i)]$ for agent $a_i$ must satisfy $p_i(0) = s(a_i)$, $p_i(T_i) = g(a_i)$, and be free of collisions. A collision is defined as either a vertex collision $p_i(t) = p_j(t)$

or an edge collision $p_i(t) = p_j(t+1) \land p_i(t+1) = p_j(t)$, where $i \neq j$. Let $P$ be the space of paths. A plan $\pi : A \mapsto P$ assigns paths to all agents. An optimal plan is one that minimizes a function $cost(\pi) = \sum_{i=0}^{M} T_i$. Finally, let "$mapf$" be an algorithm which returns an optimal plan for a problem, i.e. $\pi = mapf(G, A, s, g)$.

We introduce the concept of "obstacle switch", which corresponds to turning a (traversable) vertex into an obstacle. Let $o : V \mapsto \{0, 1\}$ be a map from vertex to an indicator variable specifying whether that vertex is turned into an obstacle (1 for obstacle, 0 for traversable space).

In this paper we introduce problems of contrastive "map-based explanation": where the goal is to identify vertices that would have to become obstacles $\{v_i \in V : o_i = 1\}$ in order for a desired path to become optimal. In other words, our goal is to identify the vertices responsible for a desired path to not be optimal, thus leading to explanations of the type "path $\pi'$ is not optimal because vertices $V^o \subset V$ are free. If vertices $V^o$ were obstacles then $\pi'$ would be optimal". This is an instance of inverse-MAPF similar to inverse single-agent shortest-paths (Ahuja and Orlin 2001; Brandao, Coles, and Magazzeni 2021). We assume the desired plan $\pi'$ is collision-free, as explaining why an infeasible plan is not optimal would be straightforward (i.e. we would only need to indicate where/when collisions happen in $\pi'$).

Let $G' = (V', E')$ be the graph obtained by applying obstacle switches $o$, i.e. $V' = \{v_i \in V : o_i = 0\}$, and $E' \subseteq E$ are the edges connecting remaining vertices $V'$. Furthermore let $f$ be the function that applies this transformation, i.e. $G' = f(G, o)$.

**Full Map-based Explanation Problem** Given graph $G$ and a desired or expected plan $\pi'$, find obstacle switches $o$ such that $cost(\pi') = cost(mapf(G', A, s, g))$ and $\sum_{i=0}^{|V|} o_i$ is minimized—in other words, find the minimal amount of obstacle switches $o$ such that $\pi'$ is an optimal plan for $G' = f(G, o)$.

**Partial Map-based Explanation Problem** Let $\pi' : A' \mapsto P$ be a *partial* MAPF plan, which assigns paths to a subset $A' \subset A$ of the agents. Given graph $G$ and a desired or expected *partial* plan $\pi'$, find obstacle switches $o$ such that $cost(constr\_mapf(G', A, s, g, \pi')) = cost(mapf(G', A, s, g))$ (i.e. such that $\pi'$ is optimal in $G'$) and $\sum_{i=0}^{|V|} o_i$ is minimized. Here $constr\_mapf(G', A, s, g, \pi')$ is an algorithm that finds an optimal MAPF solution to agents $A \setminus A'$ while constraining agents $A'$ to take paths $\pi'$.

## Methods

### Search-Based Method

One method applicable to solving both full and partial map-based explanation problems in MAPF is search-based "model-search", as used in XAIP Model Reconciliation work (Chakraborti et al. 2017). Here we propose an adaptation of model-search to these explanation problems.

Algorithm 1 shows pseudo-code for the method. The algorithm searches for a value of $o$, i.e. a set of vertices to be made into obstacles, in breadth-first order of the number of obstacles added. Each neighbor $o^n$ of state $o$ adds a single obstacle to the map, in vertex $v_j$ (line 17). Only vertices that are traversed by $\pi^* = mapf(f(G, o), A, s, g)$ are considered (line 16)—since those are the only ones that can change the optimal solution. In addition to that, without loss of completeness or optimality, we trim the search space of the problem by excluding a set of "forcefully free" vertices (line 3). For full explanation problems ForcefullyFreeVertices($\pi', s, g$) is equal to all vertices in $\pi'$, since $\pi'$ would become infeasible in case an obstacle was added to its vertices. For partial problems this is equal to the vertices of the subset of agents $A'$ in $\pi'$, as well as the start and goal vertices $\{s, g\}$ of all agents $A$—since adding obstacles here would make the explanation problem infeasible.

The search stops once $\pi'$ becomes an optimal plan for $G'$ (line 13), or if the explanation problem is infeasible (i.e. the queue is exhausted). The "$\pi'$ feasible in $G'$" condition checks whether the vertices traversed by $\pi'$ are free. The "$\pi'$ optimal in $G'$" condition then depends on whether $\pi'$ induces a full or partial problem, and it is implemented as in the respective definitions, i.e. $cost(\pi') = cost(mapf(G', A, s, g))$ for full problems and $cost(constr\_mapf(G', A, s, g, \pi')) = cost(mapf(G', A, s, g))$ for partial problems. Simply put, if $\pi'$ is feasible and its cost is the same as the cost of the optimal plan, then $\pi'$ is an optimal plan. The condition is written in this form to account for the fact that in MAPF multiple optimal plans may exist with the same cost. The method is complete since all combinations of obstacle switches are considered, and it is optimal since the queue is sorted by number of obstacles added.

---

**Algorithm 1: Search-Based InvMAPF**

---

1: **In:** graph $G$, agents $A$, start $s$, goal $g$, desired plan $\pi'$
2: **Out:** obstacle switches $o$ that make $\pi'$ optimal
3: $F \leftarrow$ ForcefullyFreeVertices($\pi', s, g$)
4: $o \leftarrow 0_{|V|}$
5: $Q \leftarrow$ PriorityQueue( )
6: $Q$.add($0, o$)
7: **while** $|Q| > 0$ **do**
8: $\quad N_{changes}, o \leftarrow Q$.pop( )
9: $\quad G' \leftarrow f(G, o)$
10: $\quad \pi^* \leftarrow mapf(G', A, s, g)$
11: $\quad$ **if** $\pi^* = \emptyset$ **then**
12: $\quad\quad$ **continue**
13: $\quad$ **if** $\pi'$ feasible in $G'$ **and** $\pi'$ optimal in $G'$ **then**
14: $\quad\quad$ **return** $o$
15: $\quad$ **for** $j$ in $\{1, ..., |V|\}$ **do**
16: $\quad\quad$ **if** $v_j \notin F$ **and** $v_j \in$ AllVertices($\pi^*$) **then**
17: $\quad\quad\quad$ $o^n \leftarrow o$ ; $\quad o_j^n \leftarrow 1$ ; $\quad N_{changes}^n \leftarrow N_{changes} + 1$
18: $\quad\quad\quad$ **if** $o^n$ **not in** $Q$ **then**
19: $\quad\quad\quad\quad$ $Q$.add($N_{changes}^n, o^n$)
20: **return** failure

---

## Incremental Optimization-Based Method

In this section we propose a fast incremental method to solve the full map-based explanation problem. It is based on the single-agent inverse shortest-paths method NISP$^{\#}$ (Brandao, Coles, and Magazzeni 2021). Our method works by incrementally obtaining a set of "bad" plans $B = [\pi_1, \pi_2, ...]$ which should be made infeasible so that a desired or expected plan $\pi'$ becomes optimal. First, the method obtains the minimal obstacle switches $o$ that makes all plans in $B$ infeasible but keeps $\pi'$ feasible. Then, it obtains a new optimal plan $\pi^*$ for $G' = f(G, o)$. If $cost(\pi') = cost(\pi^*)$ then our desired path is now optimal and the algorithm can stop. Otherwise, it adds $\pi^*$ to the set of paths $B$. The method proceeds this way until the desired plan becomes optimal (because a large amount of alternative plans was made infeasible) or until the method can no longer make progress (because it is just not possible to make $\pi'$ optimal). Pseudocode for the incremental method is shown in Algorithm 2.

---

**Algorithm 2: Incremental Optimization-Based InvMAPF**

---

1: **In:** graph $G$, agents $A$, start $s$, goal $g$, desired plan $\pi'$
2: **Out:** obstacle switches $o$ that make $\pi'$ optimal
3: $B \leftarrow \{\}$
4: **for** iter **in** 1, ..., MAX_ITER **do**
5:    success, $o \leftarrow$ *StepInvMAPF*$(A, B, \pi')$
6:    **if not** success **then**
7:      **return** failure
8:    $G' \leftarrow f(G, o)$
9:    $\pi^* \leftarrow mapf(G', A, s, g)$
10:   **if** $\pi^* = \emptyset$ **then**
11:     **return** failure
12:   **if** $\pi'$ feasible in $G'$ **and** $\pi'$ optimal in $G'$ **then**
13:     **return** $o$
14:   $B \leftarrow B \cup \{\pi^*\}$
15: **return** failure

---

The key step in the algorithm is function "*StepInvMAPF*" in line 5. This function is responsible for obtaining minimal obstacle switches $o$ that lead plans $B$ to become infeasible. The function also needs to maintain the feasibility of $\pi'$ in order to make $\pi'$ optimal for large enough $|B|$. We implement this by the following optimization problem:

$$\boxed{\textit{StepInvMAPF}(A, B, \pi'):}$$

$$\underset{o}{\text{minimize}} \quad ||o||_1 \tag{1a}$$

$$\text{s.t.} \quad o \in \{0, 1\}^{|V|} \tag{1b}$$

$$o_j = 0 \quad \forall j : v_j \in \pi'(a), a \in A \tag{1c}$$

$$\sum_{a \in A} \sum_{j : v_j \in \pi^b(a)} o_j \geq 1 \quad \forall \pi^b \in B. \tag{1d}$$

The solution to this problem will have a minimal amount of obstacles added to the graph (1a) while making each "bad" plan $\pi^b \in B$ infeasible (1d)—by placing at least one obstacle along each $\pi^b$. The problem also makes sure that no obstacles are placed along the desired plan $\pi'$ (1c), so that it is not invalidated.

For full explanation problems and a large enough MAX_ITER, the incremental method is complete because, as in the single-agent case (Brandao, Coles, and Magazzeni 2021), if a solution to the inverse problem exists then it is possible to obtain a large enough set of alternative paths $B$ that need to be made infeasible before $\pi'$ becomes optimal. At each iteration the method computes an optimal plan that is different from all $\pi^b \in B$, and adds that plan to $B$ if it is not the same cost as $\pi'$. Eventually, enough different (previously optimal) plans will be added to $B$ so that $\pi'$ becomes the next optimal plan. For full explanation problems this method is also optimal in $||o||_1$ because "*StepInvMAPF*" is optimal. If the method did not return the first solution found (line 13), then the amount of alternative plans $B$ would only increase—which could only keep or increase the value of $||o||_1$ with respect to the first solution found.

This method can also be applied to partial explanation problems, although it is not complete nor optimal in this case. To apply it here we simply need to use the appropriate optimality check in line 12, as in the search-based method, as well as constraining only the subset of agents $A'$ of the partial problem in (1c). In this case the method is not complete since it attempts to make a full MAPF-plan have lower cost than the alternatives $B$. This target plan is a single optimal plan that respects $\pi'$, even though many same-cost plans may potentially exist. In order for the method to be complete, it would therefore have to go through all optimal plans that respect $\pi'$—which is impractical in MAPF. Even so, this method may still be useful for its speed in applications where it is not essential to have optimal explanations, as we will see in the Results section. Since its computation time is negligible compared to search-based methods, it could be used to quickly obtain an explanation, before resorting to search in case it fails.

## Joint ISP Method

We finally include a third (baseline) approach to solve the map-based explanation problems. The idea of this method is to compute obstacles that simultaneously lead each of the single-agent paths $p_i$ in $\pi'$ to become optimal while ignoring other agents. We will use this method to evaluate the degree to which the single-agent inverse problem is useful in the multi-agent domain.

Let a single-agent shortest path be modeled as an LP (Ahuja, Orlin, and Magnanti 1993):

$$\min_{x \in \mathbb{R}_{0+}^{|V|}} w^\mathsf{T} x, \quad \text{s.t.} \quad Cx = b, \tag{2}$$

where $w$ is a vector of weights $w_j \in W$ which model the relative cost of movement in each edge, $x_j$ is equal to 1 if $e_j \in E$ belongs to the shortest path, and 0 otherwise. $C$ is a matrix where $C_{ij}$ is equal to 1 if $e_j$ leaves $v_i$, -1 if it arrives at $v_i$, and 0 if it does neither. Finally, $b_i$ is equal to 1 if $v_i = v_{\text{start}}$, -1 if $v_i = v_{\text{goal}}$, and 0 otherwise.

Basically, the inverse shortest-path (ISP) problem for a single agent corresponds to finding a new weight vector that leads the shortest path being the desired one $x'$. As demon-

strated by (Ahuja and Orlin 2001), can be solved as:

$$\min_{w',\psi,\lambda} \quad ||w'-w||_1 \tag{3a}$$

$$\text{s.t.} \quad \sum_i C_{ij}\psi_i = w'_j \quad \forall_{j:x'_j=1} \tag{3b}$$

$$\sum_i C_{ij}\psi_i + \lambda_j = w'_j \quad \forall_{j:x'_j=0} \tag{3c}$$

$$\psi \in \mathbb{R}^{|V|}, \, \lambda \in \mathbb{R}^{|E|} \tag{3d}$$

$$\lambda_j \geq 0 \quad \forall_{j:x'_j=0} \tag{3e}$$

$$w' \in \mathbb{R}_+^{|E|}, \tag{3f}$$

where $\psi$ and $\lambda$ are the dual variables of the constraints $Cx = b$ and $x \geq 0$, respectively, and (3b-3e) enforce the complementary slackness conditions required for $x'$ to become an optimal solution to (2). The straight-forward adaptation to our setting where obstacles can be added to the map, is to model obstacles as large costs (e.g. $w_j = 1000$ for movement over an obstacle, $w_j = 1$ for free space). Then the extension to the multi-agent setting is to simultaneously compute the ISP for each of the agents $a \in A$, while minimizing the number of obstacles added:

$$\min_{\substack{o, \\ \psi^1,...,\psi^M, \\ \lambda^1,...,\lambda^M}} \quad ||o||_1 \tag{4a}$$

$$\text{s.t.} \quad o \in \{0,1\}^{|V|} \tag{4b}$$

$$\psi^i \in \mathbb{R}^{|V|}, \, \lambda^i \in \mathbb{R}^{|E|} \quad \forall_{i\in 1,...,M} \tag{4c}$$

$$\sum_i C_{ij}\psi_i^1 = c.o_{r(j)} \quad \forall_{j:x_j^{1*}=1} \tag{4d}$$

$$\sum_i C_{ij}\psi_i^1 + \lambda_j^1 = c.o_{r(j)} \quad \forall_{j:x_j^{1*}=0} \tag{4e}$$

$$\lambda_j^1 \geq 0 \quad \forall_{j:x_j^{1*}=0} \tag{4f}$$

$$... \tag{4g}$$

$$\sum_i C_{ij}\psi_i^M = c.o_{r(j)} \quad \forall_{j:x_j^{M*}=1} \tag{4h}$$

$$\sum_i C_{ij}\psi_i^M + \lambda_j^M = c.o_{r(j)} \quad \forall_{j:x_j^{M*}=0} \tag{4i}$$

$$\lambda_j^M \geq 0 \quad \forall_{j:x_j^{M*}=0} \tag{4j}$$

where $c$ is the penalty constant (1000 in our experiments), and $r(j)$ is a map from edge index to the corresponding target vertex index. The problem is similar to (3) but has one $\psi$, $\lambda$ and $x'$ for each agent $i = 1,...,M$ (i.e. per-agent ISP), and replaces the weight vector by $o$.

This approach is not complete because it does not take interactions between agents into account. Therefore, it cannot account for situations such as when the only way to make a certain path $p'_i$ for agent $a_i$ optimal is to force another agent $a_j$ to take a path that collides with $p_i$.

The method can be used in the partial explanation setting by solving (4) over only those $\psi_i$, $\lambda_i$, $x_i$ with $i \in \{1,...,M\} : a_i \in A'$.

# Experiments

## Setup

For all experiments in this paper we used Wolfgang Hönig's[1] implementation of CBS (Sharon et al. 2015) as the MAPF

---

[1]https://github.com/whoenig/libMultiRobotPlanning

solver, and Gurobi[2] as the Mixed-Integer Linear Programming solver. Computation times were measured on a 1.90GHz x 8 Intel i7 CPU.

## Generating Benchmark Problems

We used the 8x8-grid MAPF problems from the Hönig[1] to generate a set of map-based explanation problems.

**Full Explanation Problems** For each MAPF problem we obtained an optimal plan $\pi^*$ using CBS, and then we attempted to generate a random "question" of the type "why is the optimal plan not $\pi'$?" where $\pi'$ is equal to $\pi^*$ except for two agents $a_i$ and $a_j$, $i \neq j$. The procedure we used to obtain $\pi'$ was as follows:

1. Pick two random agents $a_i$, $a_j$, $i \neq j$
2. Pick two random vertices $v_i$, $v_j$ in the map which are not in $s$, $g$, nor $\pi^*$
3. Set $\pi' \leftarrow \pi^*$ and then set $\pi'(a_i)$ equal to the shortest path through $v_i$ for agent $a_i$ (similarly for agent $a_j$)
4. If $\pi'$ is valid (no agent collisions), then attempt to solve the inverse-MAPF problem on $(G, A, s, g, \pi')$ with one of the methods (i.e. search, ISP, incremental)
5. If successful store $(G, A, s, g, \pi')$ as a full explanation problem, otherwise go to 1.

We repeated the above procedure for a maximum of 100 tries in each MAPF problem, and thus obtained three sets of problems (each set 100% solvable by one of the methods). A search/ISP/incremental method was deemed successful if it obtained a solution within a time budget of 5 minutes. This lead to the generation of a total of 100 benchmark problems.

**Partial Explanation Problems** We used the same methodology as before to generate partial-explanation problems, but where the desired plan is partial $\pi' : \{a_i, a_j\} \mapsto P$, i.e. step 3 initializes $\pi' = \emptyset$ before setting the paths for $a_i$, $a_j$. This means that all other agents are free to behave differently than they did in $\pi^*$, and it corresponds to asking a question of the type "why do agents $a_i$, $a_j$ not take paths $\pi'(a_i)$, $\pi'(a_j)$ in the optimal solution?". The process lead to a total of 140 benchmark problems.

**Problem and Method Availability** All explanation problems and methods are publicly available[3].

## Evaluation in Full Explanation Problems

Table 1 shows success rates and computation times of the three methods on the three sets of explanation problems. As expected due to completeness in the full-explanation setting, both the search and incremental methods are able to solve all problems. Joint single-agent ISP, however, is only able to solve 24 out of 36 problems generated for the incremental method, and 24 out of 39 generated for search. This shows that the single-agent inverse problem can actually be used in some MAPF problems—i.e. it is often possible to identify map changes that lead to an expected plan and avoid agent conflicts. Joint ISP is not optimal, and therefore it

---

[2]https://www.gurobi.com/
[3]https://github.com/martimbrandao/mapf-map-explanations

| Problems | Problems solved | | | Avg. computation time (s) | | |
|---|---|---|---|---|---|---|
| | Search | ISP | Incr | Search | ISP | Incr |
| rnd_incr | 36/36 | 24/36 | 36/36 | 0.61 | 2.11 | 0.55 |
| rnd_isp | 25/25 | 25/25 | 25/25 | 0.76 | 1.96 | 0.52 |
| rnd_search | 39/39 | 24/39 | 39/39 | 1.0 | 2.13 | 0.67 |
| all | 100/100 | 73/100 | 100/100 | 0.8 | 2.07 | 0.59 |

Table 1: Success rate and computation time of the three methods in three families of full explanation problems.

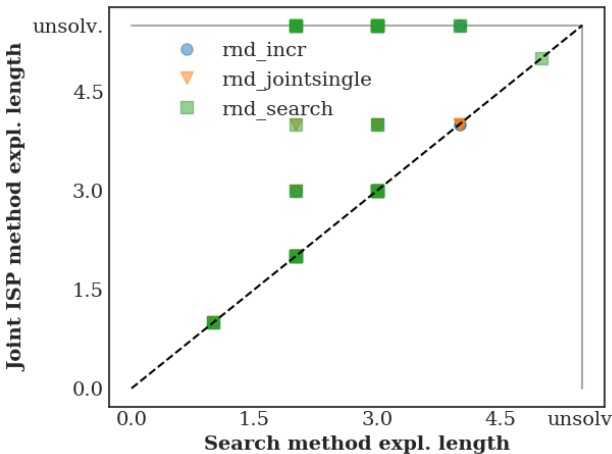

Figure 1: Explanation length of the search vs joint-ISP method, in all full explanation problems.

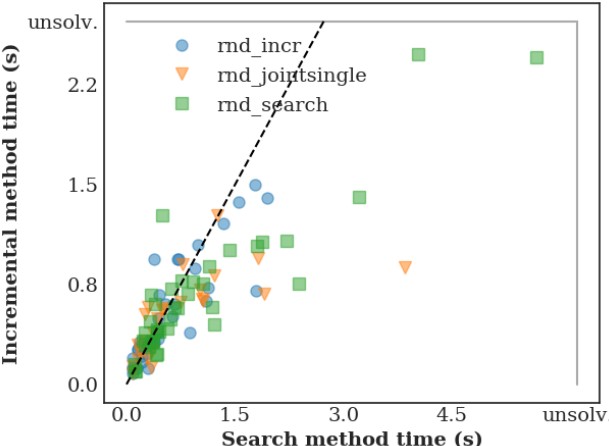

Figure 2: Computation time (s) of the search vs incremental method, in all full explanation problems. Each point is a problem. Dashed line indicates where times would be equal.

solved part of these problems with longer explanations than search/incremental. In Figure 1 we show a comparison of the explanation lengths (i.e. $|o|_1$) obtained by the joint-ISP method vs the search method—which as we have just described is often higher in ISP.

Our incremental method was slightly faster than search, as seen by the average computation times in Table 1, as well as Figure 2. The figure shows each problem as a point and compares the computation times of search vs the incremental method—showing the times are similar but often faster in the incremental method.

Figure 3 shows two examples of explanations obtained by the incremental method. First, it shows two MAPF problems with respective optimal plans (left). Then, a question of the type "why do these two agents not take the paths in purple instead (and other agents' paths stay as they are)?" (middle). For instance, the first example is basically asking why agents 2 and 3 do not avoid being close to each other and other agents' paths in the center (by going around some of the obstacles the other way). Figure 3 (right) then shows the explanations: which consist of identifying free cells that would have to be occupied in order for the desired plan to be optimal. In text, they would read "Because the cells marked in purple are free. Your desired plan would be optimal if there were obstacles on the marked cells."

## Evaluation in Partial Explanation Problems

Table 2 and Figure 4 show results on the partial explanation problems. Results for the incremental method were obtained by assuming a desired plan $\pi = constr\_mapf(G, A, s, g, \pi')$. These problems are much more challenging than full-explanation problems, as can be seen by computation times and success rates. No method was able to solve all problems. While the search method is complete and optimal, it ran on a 5 minute time budget, therefore solving only 35/41 problems generated for the incremental method and 25/46 generated for joint-ISP. As seen in Figure 4, the incremental method solved problems within 3 seconds, while the search method took up to 246 seconds (and would take longer for the problems marked as unsolved due to the time budget). We also show results obtained with a 30 minute time budget on Table 3—where search raises the success rate to 38/41 on rnd_incr and 38/46 on rnd_isp. Both tables show a column named "Collection". This consists of the sub-optimal strategy of attempting to solve an explanation problem first with the incremental method, then ISP if it fails, and only then by search. The strategy can obviously solve all problems, since each method can fully solve one of the problem sets, but at a much faster speed than if only search was used.

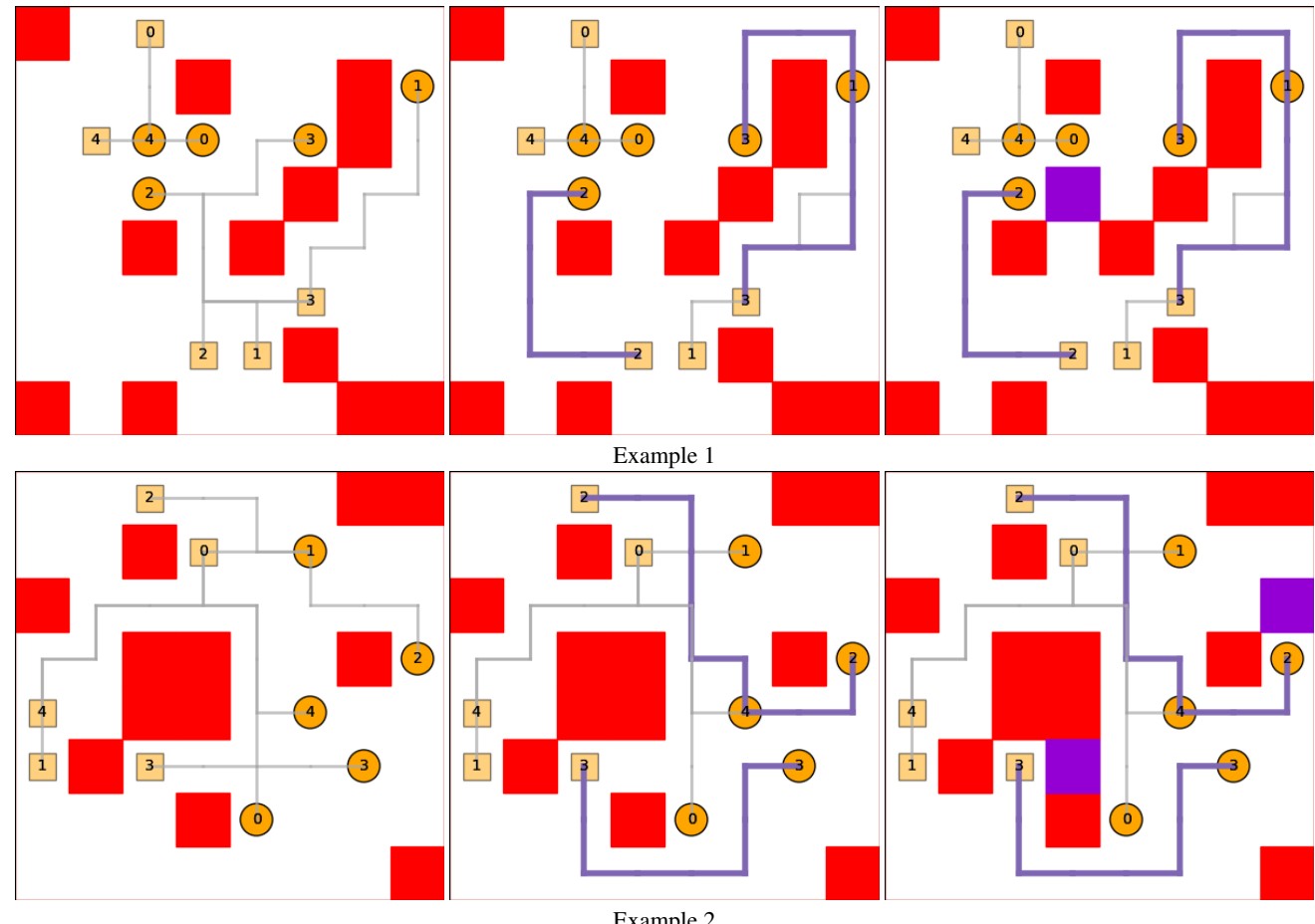

Example 1

Example 2

Figure 3: Two examples of map-based MAPF explanations. Left: a MAPF problem and plan (circles are agent start positions, squares are goals, lines are paths). Middle: the user asks "why do these two agents not take the paths in purple instead (and other agents' paths stay as they are)?". Right: the explanation "Because the cells marked in purple are free. Your desired plan would be optimal if there were obstacles on the marked cells."

## Related Work

This paper is aligned with much recent work on the topic of eXplainable AI Planning (XAIP) (Fox, Long, and Magazzeni 2017; Chakraborti et al. 2017; Sreedharan, Srivastava, and Kambhampati 2018) and explainable motion planning (Brandao et al. 2021b), in that it proposes ways to generate explanations for plans that improve users' understanding of a problem, or a method. Similarly to recent work in task planning (Sreedharan, Srivastava, and Kambhampati 2018), we compute explanations in the form of knowledge that might be missing from a user's mental model (knowledge of free space in our case), based on the user's expected plans. Since we focus on map-based explanations, our work is also inherently linked to the field of design optimization (Martins and Lambe 2013). The explanations developed by our methods could, for example, be used to guide warehouse layout designers or game map designers in improving the environments such as to obtain desired (e.g. human-predictable) paths. This is an application that has been raised as useful by industry practitioners in warehouse automation and com-

puter games (Brandao et al. 2022). In this paper we focus specifically on answering questions *contrasting* an optimal plan to one that was desired or expected—since explanations have been shown to be contrastive (Miller 2019; Lewis 1986; Lipton 1990) and recent user studies have shown MAPF experts and practitioners find these useful (Brandao et al. 2022).

Our incremental and ISP-based methods basically extend recent work on single-agent path planning explanations (Brandao, Coles, and Magazzeni 2021) to the multi-agent domain. To the best of our knowledge these are the first methods to solve map-based MAPF explanations. Few other methods have been proposed for explainable MAPF. A notable exception is the work of (Almagor and Lahijanian 2020), that proposes a method to intuitively visualize why a MAPF plan is free of collisions. A similar method has since been proposed for the continuous-motion MAPF setting as well (Kottinger, Almagor, and Lahijanian 2021).

Our incremental method is related to top-k planning (Aljazzar and Leue 2011; Katz et al. 2018), which consists of

| Problems | Problems solved | | | | Avg. computation time (s) | | | |
|---|---|---|---|---|---|---|---|---|
| | Search | ISP | Incr | Collection | Search | ISP | Incr | Collection |
| rnd_incr | 25/41 | 27/41 | 41/41 | 41/41 | 35.2 | 1.92 | 0.88 | 0.88 |
| rnd_isp | 26/46 | 46/46 | 32/46 | 46/46 | 54.93 | 2.0 | 0.88 | 1.48 |
| rnd_search | 53/53 | 24/53 | 27/53 | 53/53 | 62.35 | 1.87 | 0.84 | 36.02 |
| all | 104/140 | 97/140 | 100/140 | 140/140 | 53.97 | 1.95 | 0.87 | 14.38 |

Table 2: Success rate and computation time of the three methods in three families of partial problems, with a 5 min time budget.

| Problems | Problems solved | | | | Avg. computation time (s) | | | |
|---|---|---|---|---|---|---|---|---|
| | Search | ISP | Incr | Collection | Search | ISP | Incr | Collection |
| rnd_incr | 38/41 | 27/41 | 41/41 | 41/41 | 309.18 | 1.89 | 0.82 | 0.82 |
| rnd_isp | 38/46 | 46/46 | 32/46 | 46/46 | 343.96 | 1.92 | 0.77 | 1.38 |
| rnd_search | 53/53 | 24/53 | 27/53 | 53/53 | 61.64 | 1.88 | 0.75 | 35.47 |
| all | 129/140 | 97/140 | 100/140 | 140/140 | 217.72 | 1.9 | 0.78 | 14.12 |

Table 3: Success rate and computation time of the three methods in three families of partial problems, with 30 min time budget.

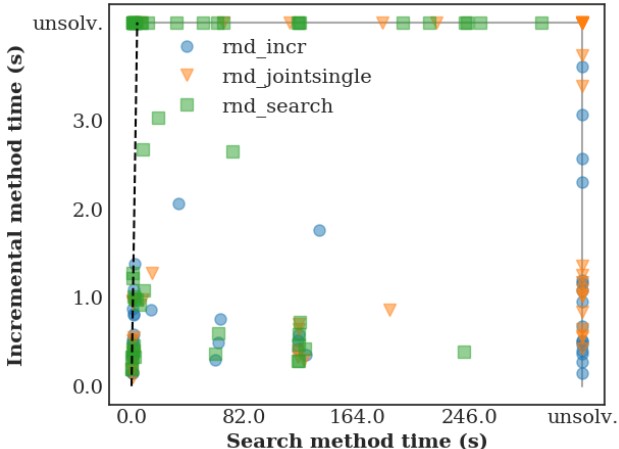

Figure 4: Computation time (s) of the search vs incremental method, in all partial explanation problems. Each point is a problem. Dashed line indicates where times would be equal.

computing the k plans with lowest cost, since we compute a set of plans with lower cost than the desired plan. However, we do not compute all plans that are lower-cost than the desired plan, but only a subset of those over which a minimum number of obstacles can be placed to obtain optimal $\pi'$.

## Conclusion

In this paper we introduced a map-based explanation problem for optimal MAPF. We formulated two variants of the problem, we proposed three methods to solve them, and we evaluated them on a benchmark that we make public for future work in the problem. The two variants correspond to answering two similar questions about a MAPF plan: 1) "why isn't this the MAPF plan?" (full-plan explanation setting), and 2) "why don't these agents take these paths instead?" (partial-plan explanation). The search-based method is op-timal in both the partial and full explanation settings. Our incremental method is very fast by focusing on incremen-tally identifying paths that need to be made infeasible in or-der for the desired path to become optimal. Even though it is incomplete in the partial setting, it is able to solve problems much faster than search, thus potentially being useful in ap-plications where optimal explanation-length is not crucial. We also showed that many problems can be solved quickly by ignoring agent conflicts in a joint single-agent ISP.

Future research directions include extensions to the con-tinuous version of these problems (Multi-Agent/Robot Mo-tion Planning), graphs with edge costs, or fast algorithms for computing the feasibility of the explanation problems. An-other interesting direction is that of conducting user studies to understand whether agent-conflict-free explanations com-puted by joint ISP are more intuitive, or more quickly under-stood than optimal explanations. User studies with profes-sional warehouse layout designers and computer game map designers would also help validate the methods—and they could provide useful information on the usefulness of this type of explanation and how to further improve it. Finally, research is needed in order to develop faster and more effec-tive algorithms for the partial-plan explanation case—both in the optimal and sub-optimal MAPF setting. These algo-rithms could leverage, for example, symmetry-reasoning (Li et al. 2021), branch-and-cut-and-price (Lam et al. 2019), or incremental optimization strategies.

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
