# OpenReview forum: "'Why Not This MAPF Plan Instead?' Contrastive Map-based Explanations for Optimal MAPF"
_icaps-conference.org/ICAPS/2022/Workshop/XAIP — XAIP 2022_

### Official Review · Reviewer_rdRw · 2022-04-28
**Review for: 'Why Not This MAPF Plan Instead?' Contrastive Map-based Explanations for Optimal MAPF**

**Rating:** 7
**Confidence:** 3

**Review:**

## Summary

The paper introduces two new kinds of MAPF explanations, namely full and partial map-based explanations, provides a benchmark of problems for such explanations, and proposes three methods for solving these problems.

The paper is an important step in exploring explanations for MAPF settings. This paper was easy to read and follow. However, there are some inherent assumptions that deserve greater discussion (more detail in feedback). The paper also lacks formal analysis like what is the bound on explanation size depending on the distance between $\pi'$ and $\pi^*$. Also, the complexity analysis of various algorithms is missing. Other than this, the remaining paper is good.

## Feedback
According to me, in its current form, the work can be improved by addressing the following issues:

1. Joint ISP assumes that the paths are collision-free and this helps it to reason about an individual agent independently of others. But this approach will only provide a lower bound on the optimal solution. The work does mention this in passing, but a greater discussion about this should be included.

2. All the benchmark problems are manhattan grid-like structures with uniform edge costs. How will these solutions extrapolate to cases where these assumptions do not hold would be interesting.

3. There is an inherent assumption that $\pi'$ can be explained by telling the explainee about connectivity, i.e., what vertices/edges are free/blocked. This will fail to handle cases where the $\pi'$ is not optimal due to collision(s). This should be noted explicitly.

4. The approach of making the k optimal solutions infeasible so that $\pi'$ becomes optimal is very similar to top-k planning. The work should comment on the similarity and differences.

5. The approach is also similar to HELM (Sreedharan et al., IJCAI 2018) which finds the smallest information missing from the user's knowledge and presents it to the user to explain the problem with a user-provided plan.

6. The benchmark only creates problems that are solvable, whereas in practice it is important to identify if a problem is not unsolvable. There should be a discussion about this.

7. How would the explanation generation time vary as we increase the number of agents whose paths are different in $\pi'$ and $\pi^*$? Currently, this number is set to 2.

---

### Official Review · Reviewer_anAK · 2022-04-29
**Very nice paper: nothing to declare!**

**Rating:** 7
**Confidence:** 3

**Review:**

The paper proposes a notion of explanation within the scope Multi-Agent Path Finding (MAPF).
In particular, the authors propose the notion of "map-based explanation": a form of contrastive explanation which exposes why a MAPF plan is not optimal, by showing where obstacles should be put in order to make a given plan optimal.
The authors propose, formalize, implement, and in-silico evaluate 3 methods to provide explanations of this sort.

Despite MAPF is not my field of expertise, I found the paper self-contained, well-written, and easy to read and understand.
The proposed notion of explanation sounds reasonable to me, and, more importantly, is clearly stated.

The proposed experiments are, to the best of my understanding, reproducible, as the authors provide the source code of their experiments.
This is good.

There are a few suggestions that, in my view, may improve future versions of the paper:
1. I guess that, in principle, the proposed notion of explanation may work for continuous spaces as well, despite being realised in (perhaps) different ways... Maybe the authors may discuss this aspect
2. the provided experiments' code is kinda tricky to test: the installation process requires a number of dependencies. May I suggest to use docker containers to automate the whole process and increase reproducibility?

---

### Meta-Review · Program_Chairs · 2022-04-30

**Recommendation:** Accept
**Confidence:** 4

**Metareview:**

The paper presents a contrastive-based explanation generation framework for explaining why certain MAPF plans are not optimal.

The paper is very well-written and it will be a great addition to the program. We suggest the authors look into the reviewers’ feedback, especially reviewer’s 2, as they provide some pointers that can further improve the quality of the paper such as elucidating on some of the assumptions made in this work.

We are looking forward to your presentation.

---

### Decision · Program_Chairs · 2022-04-30

Accept